# UAVs to Monitor and Manage Sugarcane: Integrative Review

Marcelo Rodrigues Barbosa Júnior [1,\*,†] , Bruno Rafael de Almeida Moreira [1,†] , Armando Lopes de Brito Filho [1,†] , Danilo Tedesco [1,†] , Luciano Shozo Shiratsuchi [2] and Rouverson Pereira da Silva [1]

1 Department of Engineering and Mathematical Sciences, School of Agricultural and Veterinarian Sciences, São Paulo State University (Unesp), Jaboticabal, São Paulo 14884-900, Brazil; b.moreira@unesp.br (B.R.d.A.M.); armando.brito@unesp.br (A.L.d.B.F.); danilo.tedesco@unesp.br (D.T.); rouverson.silva@unesp.br (R.P.d.S.)
2 AgCenter, School of Plant, Environmental and Soil Sciences, Louisiana State University, Baton Rouge, LA 70808, USA; lshiratsuchi@agcenter.lsu.edu
\* Correspondence: marcelo.junior@unesp.br
† These authors have contributed equally to this work.

**Abstract:** Pilotless aircraft systems will reshape our critical thinking about agriculture. Furthermore, because they can drive a transformative precision and digital farming, we authoritatively review the contemporary academic literature on UAVs from every angle imaginable for remote sensing and on-field management, particularly for sugarcane. We focus our search on the period of 2016–2021 to refer to the broadest bibliometric collection, from the emergence of the term "UAV" in the typical literature on sugarcane to the latest year of complete publication. UAVs are capable of navigating throughout the field both autonomously and semi-autonomously at the control of an assistant operator. They prove useful to remotely capture the spatial-temporal variability with pinpoint accuracy. Thereby, they can enable the stakeholder to make early-stage decisions at the right time and place, whether for mapping, re-planting, or fertilizing areas producing feedstock for food and bioenergy. Most excitingly, they are flexible. Hence, we can strategically explore them to spray active ingredients and spread entomopathogenic bioagents (e.g., *Cotesia flavipes* and *Thricrogramma* spp.) onto the field wherever they need to be in order to suppress economically relevant pests (e.g., *Diatraea saccharalis*, *Mahanarva fimbriolata*, sugarcane mosaic virus, and weeds) more precisely and environmentally responsibly than what is possible with traditional approaches (without the need to heavily traffic and touch the object). Plainly, this means that insights into ramifications of our integrative review are timely. They will provide knowledge to progress the field's prominence in operating flying machines to level up the cost-effectiveness of producing sugarcane towards solving the sector's greatest challenges ahead, such as achieving food and energy security in order to thrive in an ever-challenging world.

**Keywords:** crop-spraying aircraft systems; digital farming; meta-analysis; precision agriculture; remote sensing; *Saccharum* spp.; systematic review; unmanned aerial vehicles

## 1. Introduction

Sugarcane (*Saccharum* spp.) is key to the provision of food and energy worldwide [1]. It can massively produce sugar and lignocellulosic biomass, whether for manufacturing biofuel (e.g., bioethanol) or co-generating biopower. Hence, sugarcane proves an exciting sustainable and renewable energy crop to strategically replace fossil fuel in the Global Energy Grid [2,3]. Globally, about 200 countries are likely to produce sugarcane at industrial scale [1]. However, resource-poor farmers, harsh soil and weather, pesty outbreaking, and insufficient adoption of breakthrough plant material make it harder for them to grow it sustainably [4,5]. Development and implementation of an alternative to level up the cost-effectiveness of commercially producing sugarcane is therefore imperative and challenging. By reviewing the contemporary academic literature on engineering future-ready sugarcane, we can identify an option to endure an outperformance, namely UAV, whether for remotely monitoring and actively intervening the field [4,5].

The simplest definition of UAV describes an aircraft capable of flying both auto nomously or semi-autonomously without an on-board pilot at the command of an assistant operator near the geolocation [6]. For instance, a drone (popular terminology) is a specialty model of UAV or analogously UAS, RPV, or RPAS as official nomination [6–8]. While the original application is military surveillance mission, the contemporary universe of UAVs is multifunctional and includes, for instance, industrial-scale mining inspection [8]; rescue mission and disaster prevention for humanitarian relief (safety and health) [9,10]; forensic investigation [11]; marine fauna monitoring for ocean and coastal management [12]; habitat destruction assessment for planetary ecosystem's preservation and conservation [13]; and precision and digital agriculture for cotton, sorghum, and sugarcane [14,15] and, most frequently, cereal crops, namely barley [16], maize [17], rice [18], and wheat [19,20]. Overall, UAVs are flexible. They are likely to positively shape an extensive range of sectors in the domains of society and justice, recreation and leisure, governance, industry, business, and agriculture and environmental science, all with elementary implications into the real world.

By navigating the literature, particularly on precision and digital agriculture, we can identify relevant reviews elucidating how UAVs will likely drive thriving and responsive farming systems over the coming few years. For instance, the scoping review by Puri et al. [21] highlights the role of UAVs as disruptive aircraft systems in promoting sustainable agriculture and elaborates how they can enable farmland staffs to streamline workflows and enhance crops' productivity and quality by providing an accurate and multi-objective monitoring on the field. By further analyzing the bibliographic collection, we can examine another important systematic review by Librán-Embid et al. [22] on operating UAVs for the purpose of aligning the agriculture with changes in natural landscapes and ecosystems. If the authors' perspectives are right, we will be able to "swarm the sky" with flying robots to manage croplands, woodlands, and grasslands with more social and environmental responsiveness, while enabling safer and healthier consumers and farmers with greater profitability than possible with traditional approaches. The advancement of the domains, namely remote sensing, artificial intelligence, and robotics will likely empower stakeholders to remotely track and manage agricultural systems with greater accuracy and autonomy than ever [23]. However, hyper/multispectral sensors can make deploying UAVs into the precision and digital farming sector costly, including for sugarcane producing areas [24]. Thus, further research into low-cost sensing technology is necessary to bring the sensors closer to the customer at an accessible price [22].

A review [24] and numerous original full-text articles [4,14,15,25–27] focusing on aircraft systems for remotely monitoring and in-field management explicitly for sugarcane are available from the regular academic literature. However, to the best of our knowledge, no single study comprehensively addresses the particular applications for UAVs. Thereby, our integrative review will delve deepest into the technical–economic–social–environmental–legal ramifications of operating UAVs to monitor and manage sugarcane.

## 2. Methodology

Reviewing the specific topic of "precision and digital agriculture" requires an extensive base of research, making it challenging to define the scope and range of analysis. Thereby, we guided ourselves, screening the contemporary academic literature on aircraft systems to set out the state-of-the-art applications particularly for and elaborate on how UAVs can drive monitoring and managing sugarcane towards a thriving and responsive global sugar-energy sector.

### 2.1. Data Source and Search Strategy

We limited our search to peer-reviewed full-text articles from Elsevier's Scopus and Clarivate's Web of Science to retrieve high-visibility and impactful studies. We focused our integrative review (systematic and meta-analysis) on the period of 2016–2021 to refer to the broadest bibliographic collection, from the emergence of the term "UAV" in the typical literature on sugarcane to the latest complete year of publication in both electronic databases.

For the survey-grade queries, we elaborated research-specific strings by combining broad-researching keywords and Boolean operators, namely "sugarcane" AND "drone" OR "UAV" OR "UAS" OR "RPAS" OR "unmanned aerial vehicle" OR "unmanned aircraft system" OR "remotely piloted aircraft system". We set a list of relevant papers to identify new scholarly items to include, to broaden and strengthen our database for meta-analysis, through the "backward snowballing" technique.

Our team of peer reviewers (M.R.B.J. and B.R.A.M.) double checked the papers for readability, consistency, and eligibility by scanning through titles, highlights, abstracts, keywords, material and methods, and conclusions. They selected only studies fitting within our scope of operating UAVs to monitor and manage sugarcane, consciously excluding experiments on non-UAV platforms, duplicates to prevent bias, and any "grey literature" to the soundness of our approach. We resolved any intellectual conflict and disagreement between peer reviewers by consensus and opinion of another reviewer (A.L.B.F.). We exported our systematic research into the Microsoft Excel then organized the spreadsheets by topics, namely "progress of literature", "remote sensing", and "plant protection" to analyze it both quantitatively and qualitatively. We would like to make our database (Supplementary Materials) available to prospective readers (e.g., researchers and science policymakers) for consultation and elaboration of further investigations. Finally, we elaborated an appraisal a posteriori checklist [28,29] to systematically analyze the overall quality of research then performed our meta-analysis on "eye-catching" diagrams to picture and convey descriptive and analytical insights into ramifications of top scoring studies.

### 2.2. Quantitative and Quality Analysis

Our survey yielded 201 potential studies to include in our synthesis. However, we removed 171 studies out of the total as duplicates, clearly irrelevant upon scanning through titles, and reviews. Therefore, only 30 papers met our rigor. We quantitatively analyzed them by approaching snowballing metrics (Table 1) and based our qualitative analysis of research on methodological soundness (Table 2).

**Table 1.** Snowballing metrics for extracting relevant bibliometric information to integrative review.

| | Field Name | Description | Type |
|---|---|---|---|
| 1 | Progress of literature | Year of publication and CAGR | Numeric |
| 2 | Source | Title of journal | Alphanumeric (Text) |
| 3 | Institution | Governmental and nongovernmental organization | Alphanumeric (Text) |
| 4 | Subject | Application/purpose | Alphanumeric (Text) |
| 5 | Funding sponsor | Financing institution | Alphanumeric (Text) |
| 6 | Funding area | Field of research & development | Alphanumeric (Text) |

**Table 2.** Appraisal checklist to analyze the overall quality of research (adapted from Hanson and Jones [28] and Ogilvie et al. [29]).

| Filter | | Description | Scale |
|---|---|---|---|
| 1 | Cultivar | Did authors describe the cultivar/variety? | 1: Yes; 0: No |
| 2 | Weather | Did authors describe any weather condition? | 1: Yes; 0: No |
| 3 | Soil | Did authors characterize the soil? | 1: Yes; 0: No |
| 4 | Phenology | Did authors specify phenological stage of the sugarcane? | 1: Yes; 0: No |

**Table 2.** *Cont.*

| Filter | | Description | Scale |
|---|---|---|---|
| 5 | Season | Did authors describe the growing season or specify the date of surveying? | 1: Yes; 0: No |
| 6 | Data analysis | Did authors describe the data analytics or error metric? | 1: Yes; 0: No |
| 7 | UAV | Did authors specify the UAV? (e.g., fixed-wing or rotor-wing) | 1: Yes; 0: No |
| 8 | Sensor [a] | Did authors describe the sensor? (e.g., electro optical, LiDAR, thermal etc.) | 1: Yes; 0: No |
| 9 | Capacity [b] | Did authors describe capacity flow of spraying? | 1: Yes; 0: No |
| 10 | Calibration [a] | Did authors perform the radiometric calibration? | 1: Yes; 0: No |
| 11 | Altitude | Did authors describe the altitude of flight? | 1: Yes; 0: No |
| 12 | GSD [a] | Did authors provide information about GDS or pixel? | 1: Yes; 0: No |
| 13 | Time | Did authors describe the time of flight? | 1: Yes; 0: No |
| 14 | Overlap [a] | Did authors describe the side/front overlap? | 1: Yes; 0: No |
| 15 | Number of images [a] | Did authors specify the number of images? | 1: Yes; 0: No |
| 16 | Image processing [a] | Did authors specify the software or the processing of imagery data? | 1: Yes; 0: No |
| 17 | Velocity | Did authors describe the forward speed of flying? | 1: Yes; 0: No |
| 18 | GNSS | Did authors describe the positional accuracy? (e.g., GCPs or RTK) | 1: Yes; 0: No |
| 19 | Duration | Did authors specify the duration of the mission? | 1: Yes; 0: No |
| 20 | Accuracy | Did authors describe the statistical accuracy? | 1: Yes; 0: No |

[a] Applicable only for remote sensing. [b] Applicable only for plant protection.

## 3. Results

### 3.1. Progress of Literature: Research and Innovation

3.1.1. Historical Evolution of Publications and CAGR

By navigating the period of our integrative review, we can identify a significantly increasing number of publications over time, from just 2 in 2016 to 30 in 2021 (Figure 1), totaling 375 citations. The field grows continuously. Therefore, CAGR for the emerging yet progressing academic literature is on average 39.95%. Most notably, the FWCI is stable at 2.22, ranking it 122% above the global average research performance. Plainly, research in UAVs particularly for sugarcane can notably impact the scientific community. It focuses on remote sensing to accurately and realistically capture the spatial-temporal variability on the field, making the monitoring the largest topic cluster. By contrast, stuff-releasing and crop-spraying aircraft systems for the purpose of phytosanitization are likely to attract less intellectual interest by research leaders, science policymakers, and funders.

3.1.2. Most Active Sources

By analyzing evidence from our integrative review, we can identify 17 peer-reviewed periodicals (Figure 2). Altogether, they host relevant publications and communicate about multiple subjects and scopes from agronomy and sensors. Overall, bibliographic collection's top two journals are *Sugar Tech* and *Remote Sensing* by representing 20% and 13.33% of the total.

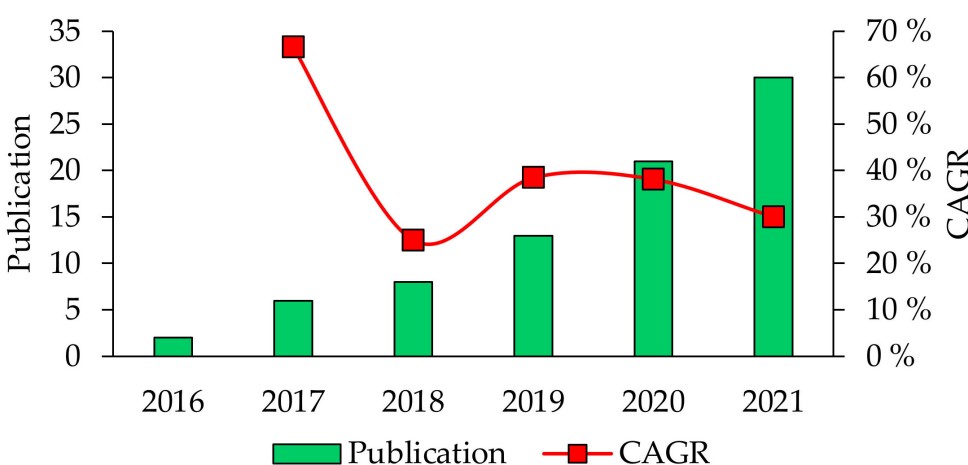

**Figure 1.** Historical evolution of research in UAVs for sugarcane.

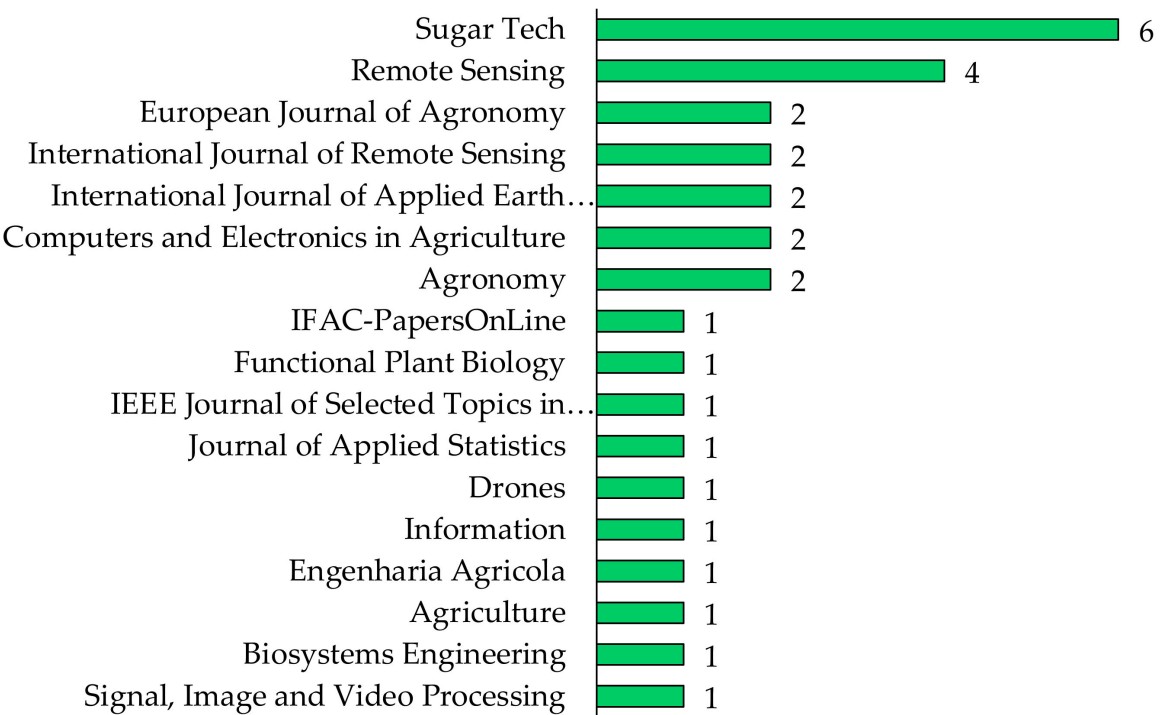

**Figure 2.** Number of relevant publications in UAVs for sugarcane by peer-reviewed periodicals.

### 3.1.3. Most Active Countries

The research into UAVs particularly for sugarcane is global and spreads across more than 10 countries (Figure 3). Most active institutional organizations are from People's Republic of China (10) and Brazil (10). Both are the world's largest producers of sugarcane. Therefore, we can draw a positive relationship between the economy and research in a timely manner. The more the country leads in producing sugarcane, the heavier it pushes funds both publicly and privately to support innovative solutions. Additionally, Thailand (05) ranks third, and the other countries are Australia (04), Japan (02), and Belgium, Canada, Nicaragua, Spain, and the USA with equal contribution (01).

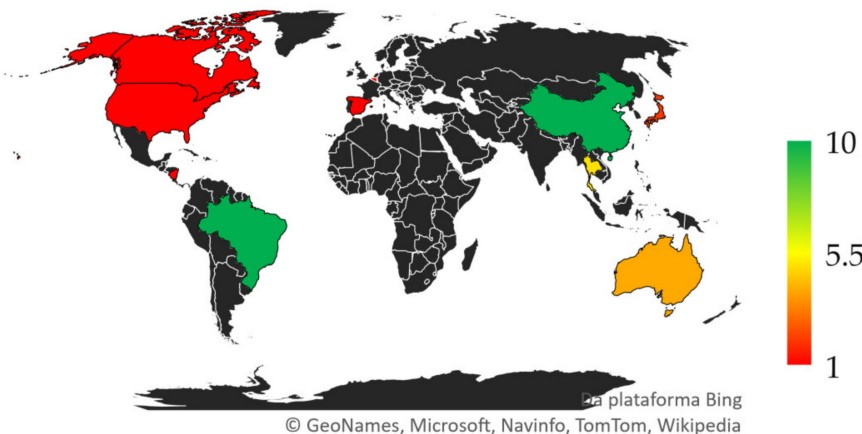

**Figure 3.** Spatial distribution of research in UAVs for sugarcane.

### 3.1.4. Most Active Subjects

The typical literature focuses most on predicting quantitative yield (Figure 4). Hence, it represents 27% of the total observation across 30 apparently eligible studies. Plant protection (pest control) and mapping of gaps' spatial expressiveness ranks second, equally contributing (17%) to the scholarly subset. Plant height determination places third in the ranking, by representing 13%. Weed detection and canopy cover fourth place in the ranking, with equal 10% of the total. By contrast, qualitative yield (Brix, Pol, fiber, CCS, and lignocellulosic composition) is the less frequent subject by representing only 7% of the total. Although the scientific community interest is the greatest on predictive analytics on yield, authors are not likely to study it qualitatively, decreasing the soundness of research.

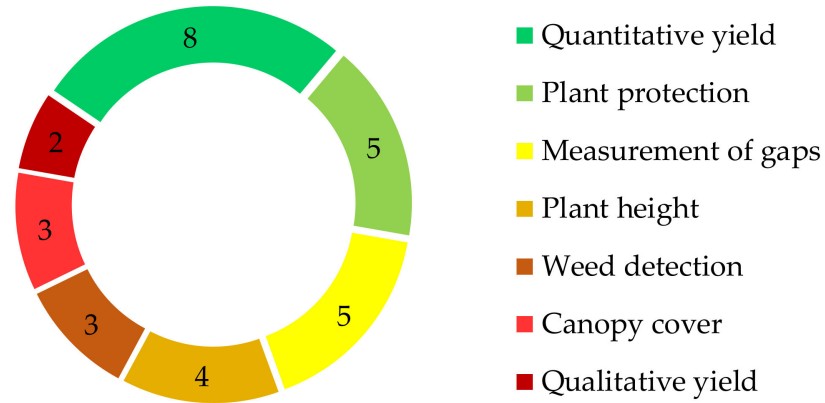

**Figure 4.** Scholarly distribution of studies in UAVs for sugarcane.

### 3.1.5. Most Active Sponsors and Funding-Absorbing Areas

Deploying UAVs into the real world is key to enabling full-scale users to successfully monitor and manage sugarcane. Until the technology gets off the academic/private ground and approaches an active commercialization, financing is necessary for updating and polishing. Examples of financing sponsors are venture capitalists, loans, companies, and scientific-politic organizations. Thereby, by delving into our integrative review, we can identify 20 studies acknowledging a funding source (Figure 5). The National Science Foundation of China represents the largest proportion of scholarly subset, contributing to 12.73% of the total and so outstripping the other funding bodies. The Coordination for the Improvement of Higher Education Personnel ranks second by representing 10.91% of the total. Guangxi Natural Science Foundation places third in the ranking by contributing 7.28% of the total.

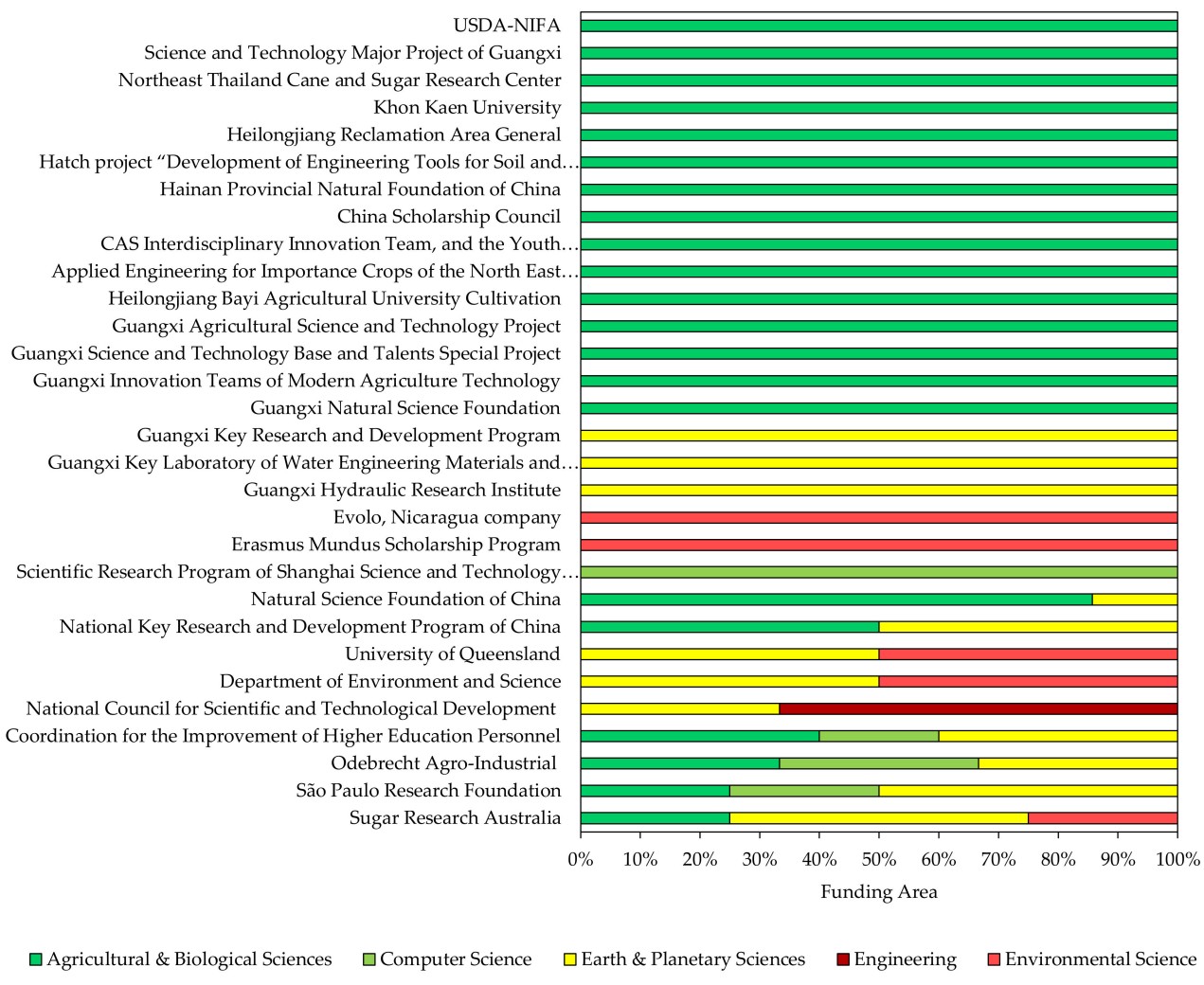

**Figure 5.** Most active financing institutions and funding-absorbing fields of research in UAVs for sugarcane.

The bibliographic collection's top funding agencies are from countries controlling the global market of sugarcane. Descriptive and analytical insights into ramifications of funding sponsors are useful to picture how they can accelerate the efforts towards research and technological development. However, we can identify inconsistencies across studies. Firstly, we introduce the dataset coverage. Of the 30 apparently eligible full-text articles, 20 provide information about financial support. Secondly, not all authors acknowledge the funding source clearly. Thirdly, funding is region specific. For instance, sponsoring is likely to widely spread across multiple official programs in the People's Republic of China, making it hard to precisely measure the attribution. Furthermore, reference to agencies in contexts not relevant to justify the funding of the research can produce false positive errors, leading to misinformation and misinterpretation. Overall, trends in funding acknowledgment can reveal priorities at the national level and the conceptual/cooperative diversity is yet immature to address the sector's greatest challenges ahead, such as developing an intricate research funding policy.

### 3.1.6. Transdisciplinarity of Research

A key tenet of research refers to integrating ideas, practices, and people from multiple disciplines. The transdisciplinary nature of science is crucial to discovery of knowledge to accelerate the efforts towards diffusing and promoting sustainable changes in society, environment, and economy. Therefore, academics and science policymakers need to tap the

research into an extensive range of disciplines if they are to effectively fulfill the emerging yet exciting innovation and technological development in UAVs explicitly for sugarcane. Thereby, by analyzing evidence from our meta-review on conceptual/collaborative diversity of transdisciplinary research, we can track frequent connections between Agricultural and Biological Science (43.24%), Earth and Planetary Science (21.62%), Computer Science (13.51%), Engineering (10.81%), Environmental Science (5.41%), Decision Science (2.70%), and Mathematical Science (2.70%) (Figure 6). The topic clusters on UAVs for sugarcane, namely remote sensing and plant protection, are likely to have stronger connections between Agriculture and Biological Science, Computer Science, and Engineering. They are thus the closest intellectual fields. By contrast, we can draw weaker linkages between Decision Science, Mathematical Science, and Environmental Science. Therefore, they are conceptually not proximate and need more collaboration by researchers, science policymakers, and funders.

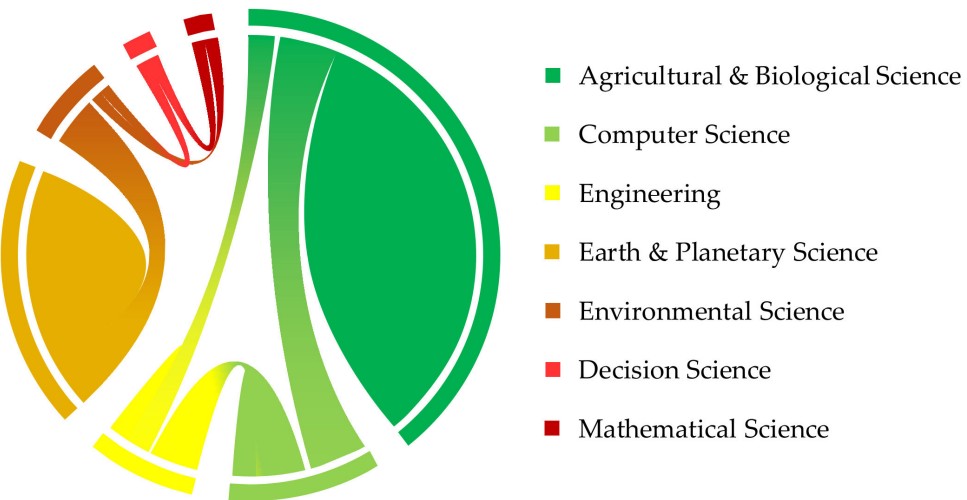

**Figure 6.** Inter-relationships between fields of research in UAVs for sugarcane.

Collaboration between organizations and institutions is an important part of developing a technology to thrive through an ever-challenging global sugar-energy ecosystem. However, we cannot find a single international collaborative action between authors from the world's leading producers of sugarcane available from our synthesis, namely Brazil, People's Republic of China, Thailand, and Australia. The trends in global-scale collaboration are not encouraging. Increasing the frequency of collaboration both nationally and internationally must therefore be a priority for future science policies. Research leaders and science policymakers must be aware of the active stance, cooperation, and coordination they need to elaborate on if they are to effectively strengthen the organizational and institutional collaboration to progress in the field's prominence in operating UAVs to monitor and manage sugarcane cost-effectively.

### 3.2. Overall Quality of Research

We could not find a single perfect study for soundness (Figure 7a). All full-text papers are missing information at any critical level, from geolocation characteristics to flight features. Scores on our database to meta-analysis can broadly range from about 21% to 94%. Therefore, typical literature can on average provide approximately 64% of the total data we might expect on a consistent methodology. Remote sensing is the largest topic cluster by representing 83.33% of the total observation across 30 studies (Figure 7b). It can meet 62% of the total 19 filters for overall quality of research. By contrast, plant protection represents 16.67% of the total and meets about 63% of the total 14 empirical criteria (Figure 7c). Therefore, well over half of the studies can score above average, irrespective of the topic cluster. Particularly on remote sensing, 4 (13.33%) out of the 25 articles can provide

85% of the total information on the UAV–imagery–sugarcane–environment relationship. The other 15% refer to missing data on weather condition and soil properties, and flight specifications such as overlap, velocity, duration, and number of images for mapping. By contrast, only 2 studies focusing on crop-spraying UAVs can score above average on quality of research in phytosanitary management. They describe the experimentation at an appreciable rate of 64–79%. Thus, the major inconsistencies refer to soil, phenology, and flight time and duration.

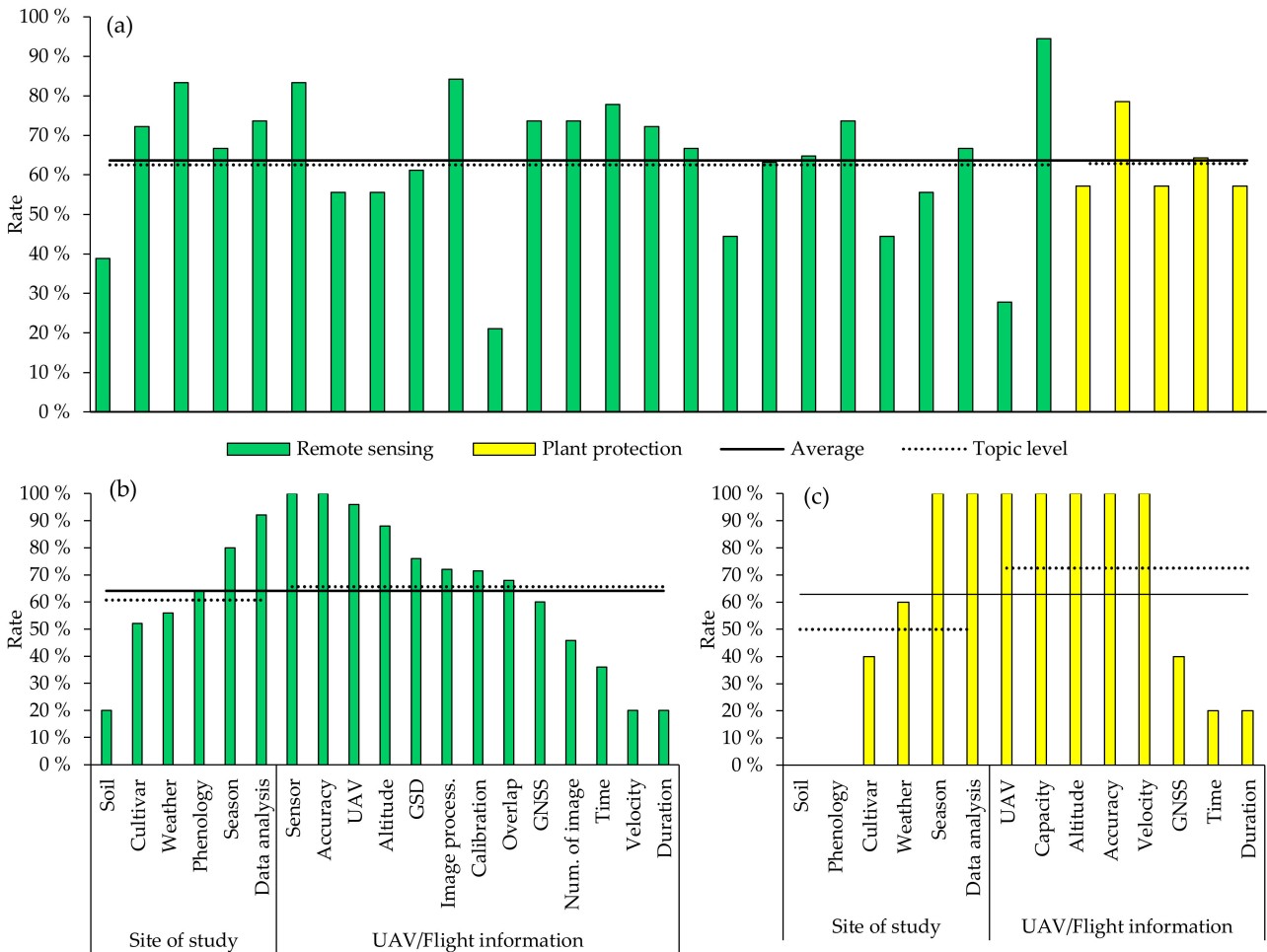

**Figure 7.** Soundness of research in UAVs for monitoring and managing sugarcane. (**a**) Overall quality for all full-text papers. (**b**) Quality of remote sensing papers by the empirical criteria. (**c**) Quality of plant protection papers by the empirical criteria.

Papers on remote sensing always describe sensor and accuracy. By contrast, authors are not likely to provide complete and trustworthy information about soil's series, velocity, and duration of the flight. They refer to them only in 20% of the articles, making them the most limiting keys. Complementarily, academics shaping the literature on plant protection are likely to provide an additional level of information in the paper. Thereby, we can fulfill approximately 50% (season; UAVs model, capacity and velocity; flight altitude, data analytics and accuracy) of the total 14 qualitative systematic criteria across 5 papers, particularly on phytosanitary management. Description on GNSS, time, and duration of the flight is less frequent (40–60% of the total studies). However, it is not less frequent than information about time and duration of the flight (20% of the studies). Soil's series and phenology do not score. They are thus the most limiting keys to the soundness of research in crop-spraying aircraft systems. Therefore, researchers should make every effort to characterize the site of study and provide information about phenology. Most

importantly, they should clearly communicate about how to operate an UAV, whether for remote sensing or phytosanitary management, to allow the study to be reproducible by an independent researcher.

### 3.3. Meta-Analytical Insights into Top Scoring Studies

By systematically checking the overall quality of research, we could extract only 17 out of the 30 potential studies to meta-analyze and set out the state-of-the-art of UAVs for sugarcane. The action of flying UAVs on sugarcane focuses on remotely monitoring and actively intervening in the field to control insets (Table 3). The top scoring studies' specialty models of UAV for precision agriculture are fixed-wing and rotary-wing. Rotary-wing is more frequent than fixed-wing, because of scale of the study and cost-effectiveness. Fixed-wing can reach remote land we might expect to not effectively reach on rotary-wing. However, it is likely more capital intensive. Other limitations refer to payload. Rotary-wing is more common. It does support optical electro camera, LiDAR, or micro-RADAR for high-quality remote sensing. Furthermore, by integrating it into precision spraying technology, we can strategically explore it in protecting the crop against insects, while simultaneously mapping the field to improve management and performance and auditing.

**Table 3.** State-of-the-art of operating UAVs to monitor and manage sugarcane.

| Cultivar | Weather | Soil Type | Phenology | Season | Statistic | Ref. |
|---|---|---|---|---|---|---|
| **Remote sensing** | | | | | | |
| N/A | Yes | Inceptisols, Entisols, and Vertisols | N/A | N/A | LDA, $R^2$, RMSE, and OA | [25] |
| RB867515 | Yes | Rhodic Hapludox and Quartzarenic Neosol | Tillering | November 2014 | SLR, $R^2$, ME, RMSE, and dr | [26] |
| N/A | N/A | N/A | Early and mid vegetative | January 2015 September 2015 | $R^2$ | [14] |
| N/A | Yes | N/A | N/A | May 2015 | Kappa | [30] |
| RB867515 | Yes | N/A | Maturity | July 2015 | ME, RMSE, and MAPE | [31] |
| Q208 | Yes | N/A | All stage | 2017–2018 | $R^2$ and RMSE | [4] |
| N/A | N/A | N/A | Early stage | 2016–2018 | ANOVA, ANCOVA, and PCA | [32] |
| N/A | Yes | N/A | Elogation and maturity | 2018 | $R^2$ | [33] |
| KK3 | Yes | N/A | Maturity | January 2018 | OA, Kappa, Person's, r, ME, MAPE, RMSE, and $R^2$ | [34] |
| N/A | Yes | Latosolic Red Soil | N/A | September 2018 | OA and Kappa | [35] |
| Liucheng-05136 | Yes | N/A | Tillering, elongation, and maturity | 2016–2017 | $R^2$, RMSE, MD, and RE | [36] |
| N/A | Yes | N/A | Maturity | November 2019 | RMSE and $R^2$ | [37] |
| KK3, K88-92, UT84-12 | N/A | N/A | Maturity | November 2018 January 2019 | ANCOVA | [38] |
| RB966928 | N/A | N/A | Tillering | 2017–2018 | RMSE | [39] |
| CTC-4 | Yes | Oxisol | Tillering | September 2019 | MAE and $R^2$ | [40] |
| **Plant protection** | | | | | | |
| GT46 | Yes | N/A | N/A | April 2018 | CV | [41] |
| GT42 | N/A | N/A | N/A | October 18 | CV | [42] |

**Table 3.** *Cont.*

| UAV | Sensor | Capacity | Calibration | Altitude | GSD | Time | Overlap | Ref. |
|---|---|---|---|---|---|---|---|---|
| | | | | **Remote sensing** | | | | |
| Fixed wing | RGB | N/A | N/A | 190 m | 4.7 cm | N/A | 90% front 50% side | [25] |
| Fixed wing | RGNIR | N/A | N/A | 150–200 m | 10 cm | N/A | 75% front 70% side | [26] |
| Multirotor | RGB | N/A | N/A | 50 m | 1.17 cm 1.14 cm | N/A | 70% front 80% side | [14] |
| Multirotor | Hyperspectral | N/A | Yes | 160 m | 11 cm | 11:50 | 60% front 30% side | [30] |
| Fixed wing | RGB | N/A | N/A | 200 m | 10.6 cm | 14:15 and 15:00 | 75% front 70% side | [31] |
| Multirotor | LiDAR Multispectral RGB | N/A | Yes | 30 and 40 m | 2–2.7 cm | 12:00 | N/A | [4] |
| Multirotor | Multispectral RGB Thermal | N/A | Yes | 60, 80, and 100 m | 3.00, 4.07, and 10.25 cm | 9:00–15:00 | 80% front 80% side | [32] |
| Multirotor | RGB Multispectral | N/A | Yes | 20 and 30 m | 4.8 cm | 10:00-14:00 | 80% front 80% side | [33] |
| Multirotor | RGB | N/A | N/A | 50 m | 2 cm | N/A | 80% front 80% side | [34] |
| Multirotor | RGB | N/A | N/A | 200 m | 90 cm | N/A | 70% front 70% side | [35] |
| Multirotor | RGB | N/A | N/A | 50 m | N/A | N/A | > 80% | [36] |
| Multirotor | LiDAR | N/A | N/A | 100 m | N/A | N/A | 30% | [37] |
| Multirotor | Multispectral | N/A | N/A | 73 m | 5 | 12:00–3:00 | 80% front 80% side | [38] |
| Multirotor | RGB | N/A | N/A | 70 m | 3 cm | N/A | 75% front 65% side | [39] |
| Multirotor | RGB | N/A | N/A | 80, 150, and 200 m | 3.5, 6.0, and 8.2 cm | 12:00 | 75% front 70% side | [40] |
| | | | | **Plant protection** | | | | |
| Multirotor | N/A | 10 L | N/A | 3 m | N/A | N/A | N/A | [41] |
| Multirotor | N/A | 10 L | N/A | 2, 3, and 4 m | N/A | N/A | N/A | [42] |
| **Image** | **Image processing** | **Duration** | **Velocity** | **GNSS** | **Accuracy** | | | **Ref.** |
| | | | | **Remote sensing** | | | | |
| 134 | PTGUI | 16 min | N/A | Metric | $R^2 = 0.9$, RMSE = 5.04, OA = 92.9% | | | [25] |
| 161 | Terra 3D | N/A | N/A | Metric | $R^2 = 0.98$, ME = −0.33, RMSE = 1.29, dr = 0.92 | | | [26] |
| 169 and 150 | Pix4D | N/A | N/A | GCPs | $R^2 = 0.61–0.97$ | | | [14] |
| 113 | ERDAS LPS | N/A | N/A | GCPs | OA = 92.5%, Kappa = 0.87 | | | [30] |
| 128 and 146 | Terra 3D | N/A | N/A | Metric | ME = 0.08 m, RMSE = 0.40 m, MAPE = 6.66 | | | [31] |
| N/A | Pix4D | 20–25 min | 4.4 m/s | GCPs | $R^2 = 0.92$, RMSE = 0.009–0.046 m | | | [4] |
| N/A | Pix4D | 15 min | N/A | GCPs | GCV = 4.54–13% | | | [32] |
| N/A | N/A | N/A | 2.2 m/s | GCPs | $R^2 = 0.30–0.88$ | | | [33] |
| 758 and 723 | Metashape | N/A | N/A | GCPs | r = 0.95, ME = 0.10, MAPE = 4.56, RMSE = 0.16 | | | [34] |

**Table 3.** *Cont.*

| Image | Image processing | Duration | Velocity | GNSS | Accuracy | Ref. |
|-------|-----------------|----------|----------|------|----------|------|
| | | | | **Remote sensing** | | |
| 236 | Context Master | N/A | N/A | Metric | OA = 94% | [35] |
| N/A | PhotoScan | 12 min | N/A | N/A | $R^2$ = 0.15–0.96, RMSE = 1.09–18.32 | [36] |
| N/A | R | N/A | 1 m/s | N/A | $R^2$ = 0.97, RMSE = 1.33 kg m$^2$ | [37] |
| N/A | Pix4D | N/A | 6 m/s | GCPs | $R^2$ = 0.91 | [38] |
| N/A | PhotoScan | N/A | N/A | GCPs | RMSE < 0.18 m | [39] |
| N/A | Metashape | < 60 min | 7 m/s | Metric | $R^2$ = 0.97, MAE = 0.02 m | [40] |
| | | | | **Plant Protection** | | |
| N/A | N/A | 2:43 min | 4 m/s | RTK | CV = 11.59–32.34% | [41] |
| N/A | N/A | N/A | 4, 5, and 6 m/s | RTK | CV = 0.15% | [42] |

ANCOVA: covariance analysis; ANOVA: analysis of variance; CV: coefficient of variation; dr: enhanced Willmott concordance coefficient; GCPs: ground control points; Kappa: Kappa coefficient; LDA: linear discriminant analysis; MAE: mean absolute error; MAPE: mean absolute percentage error; MD: mean derivation; ME: mean error; OA: overall accuracy; PCA: principal components analysis; r: Person's correlation; $R^2$: coefficient of determination; RMSE: root mean square error; RTK: real-time kinematic; SLR: simple linear regression. N/A indicates not applicable or not available for the data.

UAVs are flexible and fit easily anywhere, regardless of the technological level. Thereby, they enable monitoring and managing an extensive range of genotypes, such as the Brazilian RB867515 and the Chinese Thai Khon Kaen 3 (KK3). Hyper/multispectral sensors onboard an aircraft system can perform an accurate remote mapping. They acquire spatial data at a pinpoint spatial resolution, ranging from 1 to 10 cm. However, they can be costly to deploy and often require an expert for calibration to prevent any radiometric inconsistency. Radiometric calibration, by converting raw imagery data from a sensor into a common physical scale upon the reflectance, can improve performance through the mission, even if atmospheric interference is expressive. However, authors do neither perform nor describe it clearly. Another relevant feature is geometric positioning, and the authors rely on GCPs and GNSS-RTK to perform a mission with accuracy and consistency, from takeoff to landing. Both provide georeferencing with centimeter-level precision. Comparatively, GCPs is economically more appealing than GNSS-RTK. However, GNSS-RTK is more appropriate to address instances such as 3D terrain evaluation.

If the mission we want to perform is remote sensing, whether for scouting or surveying, then we need to decide on how we will perform the processing of image. Furthermore, by delving into analytical ramifications of top scoring studies, we can identify AgiSoft Metashape (St. Peterburg, Russian) and Pix4D (Prilly, Switzerland) as the most frequent photogrammetric applications. However, both can be costly to deploy, driving the handling of alternative software or open-sourcing programming languages for statistical computing and graphics. For instance, Python's pipelines and R's packages and APIs already include functions for processing geographical data. Both enable the user to track accuracy/precision on metrics such as $R^2$, RMSE, and Kappa coefficient.

## 4. Discussion

### 4.1. Insights into Scholarly Ramifications of our Integrative Review on UAVs for Sugarcane

A swarm of aircraft systems is arising in the agriculture's sky. Furthermore, because they are likely to shine through the precision and digital farming sector, we might expect UAVs as disruptive aerial technological platforms to radically change the future of producing specialty cash crops. Our integrative review synthesizes and analyzes the contemporary academic literature on UAVs particularly for sugarcane. Research and development is modest yet provides forward knowledge, so we can elaborate on how UAVs can enhance monitoring and managing sugarcane-farming frameworks. UAVs are emergent yet ap-

pealing programmable, addressable device-to-system solutions for scouting, surveying, and actively intervening in the sugarcane producing areas. They do support high-quality sensors (e.g., RGB [4,14,25,31–36,39,40], multispectral [4,32,33,38], hyperspectral [30], Li-DAR [4,37], and thermal [32]) to capture the spatial-temporal variability of the field with pinpoint accuracy, flexibly and realistically (3D ortho-mosaic model). Hence, they can perform exceptional remote sensing and provide useful imagery data on canopy, enabling stakeholders to make early-stage decisions at the right time and place, whether for mapping, re-planting, fertilizing [40], or protection [41,42], more precisely than is possible with traditional approaches. UAVs with precision spraying technology can effectively control insects, while preventing the user from overspending resources and the environment from contamination/pollution. They do not need heavy trafficking or touching of the object as we might expect for operating ground-level sprayer equipment [25,26]. Most notably, they can promote occupational safety and health by preventing workers from experiencing a direct exposure to harmful chemicals during manually spraying on field. Remote sensing is the foundation of operating UAVs to monitor sugarcane. By contrast, crop-spraying technology is yet at an early stage of development. Therefore, the scientific community must collaborate if it is to effectively advance the research and development in aircraft systems for the purpose of airborne phytosanitization. Another direction for researchers is to engineer UAVs for releasing natural enemies of insects and ripening to broaden the range of functions to address an increasing pressure to develop a multi-objective management.

By analyzing our insights into ramifications of plant protection, we can find neither a single reference to an aircraft system capable of spreading entomopathogenic agents on field to control insects nor relevant evidence on ripening. However, by approaching altimetric data from reputable websites reporting on the business of UAVs, we can screen out companies innovating in marketplaces of aircraft systems for biological control and ripening. A company [43] pioneers the technological development in UAVs for releasing biological agents to control pests in commercially relevant crops. It develops and delivers stuff-releasing UAVs to position *Cotesia flavipes* and *Thricrogramma* spp. in precise points of the field, needing the action by endoparasites to suppress the target-pest biologically rather than chemically. The company grows eggs of an organism and *Thricrogramma* spp. together for parasitic infection and spreads them in bulk onto sugarcane through a rotary dispenser module onboard of the aerial platform. The system is precise and operationally effective by working 350 ha d$^{-1}$ per UAV. The operational efficiency for *C. flavipes* is 250 ha d$^{-1}$ per UAV, and the system releases it through biodegradable recipients. The company feeds field-level data back to customers to optimize managing and offers them professional transportation, operations, servicing, and maintenance of the equipment, preventing them from overspending time upon making trivial decisions. An additional level of information includes farm-to-industry comparative analytics, check-up of every field for performance and auditing, inspection of pack and reject for waste management, and testing and training of UAVs and staff to ensure consistent operations. The global sugarcane system is facing labor shortage and spiraling expense. Furthermore, because the crop is in danger of returning to the days when pesty outbreaking could make it arduous for farmers to produce high-quality feedstock, whether for food and biofuel, the company's vision is to promote biological control of pests by creating "smart aircraft systems" to optimize and reimagine routine management, safeguard the commercial future of producers, and secure the access of industry to an affordable high-quality material.

Another company [44] develops, delivers, and services UAVs, robots, autopilot, artificial intelligence, and IoT across more than 40 countries and regions. The startup's crop-sprayer UAVs enable every user to intervene in the field safely and effectively through simple yet intelligent operation. The fully autonomous aerial system is capable of spraying liquid onto every needing crop, for instance, sugarcane for ripening with uniformity and pinpoint accuracy. It leverages a cutting-edge intelligent rotary atomizing module to spray droplets downdraft wherever they need to be, whether for protecting or ripening the crop without significant drifting. Hence, every facility secures quality, vastly reduces waste, and

protects the environment by cutting 30% of the application of pesticide. The technology provides flight speed (12 m s$^{-1}$) and spraying flow (5.6 L min$^{-1}$) sufficient to achieve an efficiency of 14 ha h$^{-1}$ or greater, since a worker can operate up to five UAVs simultaneously. It can potentially save the work of 100 people and solve the problem of an ageing rural population. Flying an aircraft system can compensate for the limitation of ripening by ground-level sprayer equipment. Sugarcane is architecturally large and adds natural physic mechanical barriers for a tractor to operate across a dense, actively growing vegetation. Even if equipment is an airplane or helicopter, spraying pesticide or ripener onto the canopy is challenging. Both can only intervene the sugarcane at late-stage condition, so the farmer cannot intervene cost-effectively. Since every transformation in the world begins with an idea, drone-as-a-service offering companies can catalyze reimagining the global sugar-energy ecosystem to thrive through an ever-challenging world. Furthermore, the scientific community can be an ally in progressing in the field's prominence in operating flying machines to remake the farm-business universe of sugarcane and overcome the industry's challenges ahead.

### 4.2. Limitations and the Ways Forward

UAVs prove useful to enhance monitoring and managing sugarcane. However, they are likely to have techno–economic, phenotype–environmental, and social–ethical–politic limitations. Furthermore, by analyzing relevant evidence from our integrative review, we can identify gaps we need to fill if we want to advance the technology over the coming few years of research and development by a set of strategic, proactive, catalytic, and capacity-building actions.

#### 4.2.1. Techno–Economic and Autonomy

An immediate technical limiting factor is low endurance. Since a battery's nominal lifespan is often short, mainstream UAVs do not perform an autonomous or semi-autonomous mission longer than 1 h. An emerging yet exciting solution to improve endurance is hybrid power system. It leverages lower-energy combustion engine and fuel cell to supplement an arrangement of LiPo, enabling it to produce and store current longer than is possible with a conventional battery. Hence, it considerably extends the flight to 3 h, making an aircraft system scalable rather than unfeasible to scout/survey throughout an extensive landscape [45]. However, it can be costly to deploy and does not compensate for the platform's sensitivity to stress. For instance, precipitation and pollution can negatively impact both quality of an optical sensor and effectiveness of communication and navigation. Another techno–economic limiting factor refers to dependence on payload. High-quality electro optical and multi/hyperspectral camera, LiDAR, and thermal and other similar payloads are capable of effectively collecting a massive volume of imagery data. They are key to the communication system succeeding. However, they can make it costly and demand an appreciable amount of space to fit onboard an aerial platform. Compactness is another relevant endurance feature we need to decide about for safe-by-design upgrading [45]. UAVs can assist in enhancing full-scale production of sugarcane. However, they can demand an assistant operator to perform a mission, from takeoff to landing. Semi-autonomous monitoring or management is rather complex and often makes the project costly and does not provide sufficient accuracy and flexibility for precision and digital agriculture. Plainly, autonomy is crucial to minimizing operator's workflow and control risks of delay, overload, and fatigue. Development of fully autonomous UAVs is challenging, although domains of remote sensing, robotics, and artificial intelligence are progressing and have become more agile and faster than ever. A direction of future research and development for the progression of autonomy is therefore to engineer lighter yet more power-effective flying machines [46].

### 4.2.2. Phenotype–Environmental

Sugarcane is a perennial grass crop. As it grows and develops, vegetation becomes denser and make it harder for an autonomous aircraft system to look through the objective area to, for instance, remotely detect gaps for re-planting or fertilizing [40]. Therefore, timing the flight is crucial to monitor the field and capture the pattern with sufficient accuracy to support making precise decisions. Even if imagery resolution is high, flying a UAV at the transitory tillering-elongation stage is not likely to accurately image the spatial representation of the gap. The plant is sufficiently large to architecturally overlap the target, making it "invisible" rather than "achievable" from the platform. Fortunately, an optimal combination exists between flight's altitude and phenological condition for the purpose of mapping gaps with pinpoint accuracy. However, further in-depth economic analysis is necessary for validation to scale [40]. Additionally, an actively growing vegetation denser than permeable often makes it harder to spray pesticides and release natural enemies wherever they need to be, whether for successfully controlling insects (e.g., *D. saccharalis* and *M. fimbriolata*), diseases (e.g., sugarcane mosaic virus), or weeds. Even if the target is visible in the range of action, an autonomous crop-protecting aircraft system is not likely to perform as we might expect on an optimal architectural condition, without critical barriers to the workflow. Another way the environment can negatively impact the UAV's performance is by deteriorating the quality of systematic communication or navigation. Since UAV is dependent on a radio line of sight, an obstacle naturally existing in the agroecosystem can interfere with the trafficking and receiving of 900 GHz, 2.4 GHz, or 5.8 GHz as an active frequency for command, control, and action. Setting up Wi-Fi in the work-up area can compensate for the limitation of flying a low-frequency aircraft system on challenging landscape, reducing poor communication and navigation [45]. Another simple yet effective strategy is to properly define the location to perform takeoff and landing.

### 4.2.3. Social–Ethical–Political

By reviewing the period of our meta-analysis, we can identify a progressing business horizon at the world's commercial UAV market, from USD 2 billion in 2016 to about USD 127 billion in 2020 [47]. While the technological and entrepreneurial co-development is evolving, the regulation is lagging [48,49]. A factor delaying the progress in the field's prominence of regulating both commercial and private application of UAVs is the open-source development of the technology. An open-source technology is flexible and often makes it challenging for policymaking and holistically governing the implementation and trade-offs [48]. Thereby, experts in technology in society are likely to inform the policymakers and governments of the commitment, cooperation, and coordination they need to elaborate on if they are to effectively promote and perpetuate UAVs in agriculture with proper procedures and guidelines [48–50]. Entities must be based on safety, legality, privacy, informational integrity, and human–machine divide to set out ethical standards to communicate about the moral values and provide a reference for operating pilot-agnostic aircrafts systems transparently, safely, and fairly [48].

The structure of the global sugar-energy sector is heterogeneous, making it hard for official regulatory authorities to effectively oversee and govern the applications and implications of UAVs in agricultural systems focusing on food and energy. However, they must take an active stance in broadcasting relevant information between stakeholders, whether by verbal, analogue (e.g., television), or digital (e.g., computer and mobile device) media [51]. Effective communication is likely to be based on awareness of the flying. While the hobby flight is legal, the commercial flight is not legal and requires extensive progress towards achieving the government's objectives [45–50], although North American, European, and Australian social–ethical policies are already mature [48]. If our meta-analytical perspectives are right, commercial UAVs will empower farmers to streamline remote intervention and make analytical decisions on management at the right time and place, whether for mapping, re-planting, fertilization, or protection. However, institutional (i.e., the scientific community) and governmental entities need to develop a particular

regulatory regime and provide a framework to assess, for instance, the threats to operators, farmers, and drone-delivery and servicing companies.

"Privacy vs. public" and "human vs. machine" are dilemmas we need to debate about for disambiguation and progress on safely operating UAVs. They can impact both positively and negatively the commercial exploitation of UAVs by farmers or companies. Private companies can provide professional servicing and an additional level of actionable information and interaction with an expert to any assistance both remotely and in situ, empowering the contractor to optimally explore the technology. However, governments must commit to promoting the technology by programming and providing incentives to low-income producers, especially in zones where agricultural arrangements are not equitable and vulnerable. Clearly, any governmental incentive will require absorbing a portion of the financial risk. However, low-income producers who cannot push ownership funds for affording an aircraft system will be able to benefit from the technology with equitability and competitiveness at the market. At the boundaries of "human vs. machine", operators can remotely control commercial UAVs from a safe location, from takeoff to landing [50]. Large-scale staffs usually are knowledgeable about how to operate an aircraft system properly, avoiding unsafe missions. Plainly, training is instrumental to operate UAVs. Furthermore, because it can be an enabler for constructing a safe human–machine divide, sugar-energy companies must make it available for workers to exercise specific skills to competently perform operations. The government could professionalize low-income producers to safely fly UAVs by scheduling and conducting public courses. Every stakeholder in the sugar-energy ecosystem must have access to the technology for justice. If not, it will develop unilateral technological dynamism and consequently the monopolization will arise from the ecosystem [52].

## 5. Conclusions

By analyzing evidence from our integrative review, we can draw the following as concluding remarks and outlooks: UAVs prove useful to sugarcane; they are emergent yet exciting high-throughput aircraft systems to monitor and manage by navigating through the field to scout, survey and actively intervene, both autonomously and semi-autonomously; remote sensing is the foundation of operating UAVs. Furthermore, by integrating them to precision spraying technology, we can perform another instrumental function to routine management, namely crop protection. At the boundaries of remote sensing, UAVs do support high-quality sensors, such as hyper/multispectral optical camera, LiDAR, and thermal. Hence, they capture the spatial-temporal variability of the field with greater accuracy, flexibility, autonomy, and realism than possible with orbital data-acquisition platforms and ground-level equipment. At the boundaries of plant protection, they can leverage cutting-edge intelligent spraying modules to drop liquids onto needing areas, whether for controlling insects with plant-level (stalk and leaf) or soil-level precision. Thereby, they can empower farmers to streamline workflows, save on arduous labor of farmland staff and, most notably, address the increasing pressure to reduce spraying chemicals and promote occupational safety and health. Research and development in UAVs for sugarcane is continuously increasing, yet it is at an early stage of progress. Therefore, researchers and science policymakers must take an active stance and collaborate to get the technology off the academic ground if they are to effectively promote and perpetuate it into the real world. UAVs will shine through precision and digital agriculture sector over the coming few years. They will act as an opening of disruptive solutions to radically change our thinking and contribute to sustainably producing sugarcane as a specialty crop for food and energy. Thereby, the scientific community, policymakers, and governments must be aware of the commitment, cooperation, and coordination they need to elaborate on if they are to effectively address the technology with transparency, safety, and justice for every stakeholder in the global-scale sugar-energy ecosystem. A direction to progress in the field's prominence in operating flying machines to monitor and manage sugarcane is to engineer lighter yet more power-effective UAVs to improve endurance and autonomy.

Another is to tailor today's engineering models to release items whether for biological control of pests, ripening, or weeding. If our descriptive and analytical perspectives are right, multi-objective aerial platforms will succeed more in sustaining a cost-effective management by solving labor shortage and spiraling expense of an ageing rural population in the sector. Thirdly, no single study could even remotely focus on imaging sugarcane to capture information about quality, whether for Brix, sucrose, or fiber. Therefore, further in-depth investigation could hypothesize whether an aircraft system with state-of-the-art sensor is capable or not of tracking "sugary transpiration" throughout the cycle. If yes, it could empower the farmer to harvest the quality of the crop for fine-scale industrialization. Finally, policymaking must include every farmer in the social–ethical development to bring about an equitable, fair, and stable arrangement, capable of thriving through a challenging politic-scientific scenario ahead.

**Supplementary Materials:** The following supporting information can be downloaded at: https://www.mdpi.com/article/10.3390/agronomy12030661/s1, supplementary File S1: Table S1. List of meta-analysis publications.xlsx.

**Author Contributions:** Conceptualization, M.R.B.J. and B.R.d.A.M.; methodology, M.R.B.J., B.R.d.A.M.; validation, M.R.B.J. and B.R.d.A.M.; formal analysis, M.R.B.J., B.R.d.A.M. and A.L.d.B.F.; data curation, M.R.B.J. and B.R.d.A.M.; writing—original draft preparation, M.R.B.J. and B.R.d.A.M.; writing—review and editing, M.R.B.J., B.R.d.A.M., A.L.d.B.F., D.T., L.S.S. and R.P.d.S.; visualization, M.R.B.J., B.R.d.A.M., A.L.d.B.F., D.T., L.S.S. and R.P.d.S.; supervision, R.P.d.S.; project administration, R.P.d.S. All authors have read and agreed to the published version of the manuscript.

**Funding:** We acknowledge support for the publication of this work by the Publishing Fund of Graduate Program in Agronomy (Soil Sciences).

**Data Availability Statement:** Data sharing not applicable. No new data were created or analyzed in this study. Data sharing is not applicable to this article.

**Acknowledgments:** We would like to acknowledge the Coordination for the Improvement of Higher Education Personnel (CAPES) for the financial support (code 001) to the first author and the Laboratory of Machinery and Agricultural Mechanization (LAMMA) of the Department of Engineering and Mathematical Sciences for the infrastructural support.

**Conflicts of Interest:** The authors declare no conflict of interest.

## Abbreviations

| | |
|---|---|
| CAGR | compound annual growth rate |
| FWCI | field-weighted citation impact |
| GCPs | ground control points |
| GNSS | global navigation satellite systems |
| GSD | ground sample distance |
| RPA | remotely piloted aircraft |
| RPAS | remotely piloted aircraft system |
| RPV | remotely piloted vehicle |
| RTK | real-time kinematic |
| UAS | unmanned aerial system |
| UAV | unmanned aerial vehicle |

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
