# Peer review of "UAVs to Monitor and Manage Sugarcane: Integrative Review"

_agronomy, doi:10.3390/agronomy12030661_

Round 1

Reviewer 1 Report

Manuscript Number: agronomy-1605914

Title: UAVs to Monitor and Manage Sugarcane: Integrative Review

This is a prospective study that proposed an interesting review that focus on the technical-economic-social-environmental-legal ramifications of operating UAVs to monitor and manage sugarcane. The manuscript is overall clearly written and well presented. The results appear to be valid and the methodology is appropriate. However, there are some concerns need to be addressed before reviewer is able to recommend this manuscript for publication.

L84-85: This statement is confusing. The hyper/multispectral sensors used on UAVs are generally different from those used on ground machinery, the size, performance and cost should be different.

L113: What do abbreviations stand for: ”M.R.B.J. and R.R.A.M.“?Same as for L120 "A.L.B.F." .

L132: it is unclear if the number of literatures (total 30 papers) analyzed is sufficient to conduct the quantitative analysis? Please rationale this issue.

L237: How the quality of research was evaluated? It is unclear how the scores were determined? Please clarify this.

L258-259: The statement is confusing. What is the "opposite"? Please revise.

L292: What is "radiometric calibration"? Please explain.

Reviewer 2 Report

The Review is an very nice work in relation to the actual and future possibilities of UAVs to assist sugar cane production. 

Small corrections I found 

line 479 delete first "requires"

line 482 change "perspectives right" by perspectives are right

line 505 relace "professional" by professionalize

line 541 replace de point at the end of the line by a semicolon

Line 546 change "perspectives right" by perspectives are right

Line 553 delete the point at the end of the line

Reviewer 3 Report

The authors wrote a scoping review on uav potentials for the sugarcane industry. The methodology and results are well-described. However, the abstract and discussion need to be improved in order to reflect the title of the review. 

Detailed comments are as follows:

Abstract: abstract does not highlight/summarize the review content e.g. the objective, the methodology, main findings are practicality of the findings.

Lines 49-51: please add references to support this claim, and add more detailed discussion on this matter.

Line 49: 'low farmer's cost-effectiveness' - the term needs to be explained/rephrase, the meaning is not clear. 

Line 80: 'If authorship perspectives and rights' -  the term needs to be explained/rephrase, the meaning is not clear. 

Discussion: in general, the discussion barely focus on the findings of the review and on sugarcane crop industry. it is suggested that the discussion is improved to throughly demonstrate uav potentials on sugarcane industry.

Section 4.1: there are no reference at all for these comments. although it describes the general benefits of uav for agriculture, these benefits must have been observed from other studies, and therefore should be cited. 

Section 4.2: the discussion for this subsection should be more detailed by referring to the reviewed literature.

Reviewer 4 Report

I have gone through the article titled 'UAVs to Monitor and Manage Sugarcane: Integrative Review' and found it an interesting piece of work. I think the development of crop spraying aircraft systems to transform traditional agriculture into digital one would be a step forward towards advancement in crop yields. The paper is written nicely the appropriate literature is reviewed. However, I feel the following comments will be helpful in further improvement:

1) I feel abstract is more towards general description. More focus towards sugarcane specifically is required

2) Authors should the limitations a bit more in detail (section 4.2)
